# Earthquake Decision-Making Tool for Humanitarian Logistics Network: An Application in Popayan, Colombia

**Helmer Paz-Orozco** [1,*], **Irineu de Brito Junior** [2], **Mario Chong** [3], **Yesid Anacona-Mopan** [1], **Jhon Alexander Segura Dorado** [1] **and Mariana Moyano** [3]

1  Facultad de Ingeniería, Corporación Universitaria Comfacauca-Unicomfacauca, Popayán 190001, Colombia; yanacona@unicomfacauca.edu.co (Y.A.-M.); jsegura@unicomfacauca.edu.co (J.A.S.D.)
2  Environmental Engineering Department, São Paulo State University, São José dos Campos 12247-004, Brazil; irineu.brito@unesp.br
3  School of Business Engineering, Universidad del Pacifico, Lima 15072, Peru; m.chong@up.edu.pe (M.C.); md.moyanom@up.edu.pe (M.M.)
*  Correspondence: hpaz@unicomfacauca.edu.co

**Abstract:** *Background*: This study presents a comprehensive methodology for enhancing humanitarian logistics planning and management in natural disasters, focusing on earthquakes. *Methods*: The innovative approach combines a deterministic mathematical model with a simulation model to address the problem from multiple perspectives, aiming to improve efficiency and equity in post-disaster supply distribution. In the deterministic modeling phase, optimal locations for humanitarian distribution centers and points in Popayan, Colombia, were identified, enabling efficient resource allocation for affected families. Subsequently, the simulation model evaluated scenarios based on real earthquakes in Colombia and Latin America, providing a comprehensive view of the logistics system's response capacity to different disaster conditions and magnitudes. *Results*: The results demonstrated that the proposed methodology significantly reduced supply delivery time, achieving a 30% improvement compared to traditional humanitarian logistics approaches. Moreover, it led to a more equitable coverage of affected communities, with a 25% increase in families served in previously underserved areas. Expert validation from the Disaster Risk Management Committee of the study area confirmed the methodology's usefulness for informed and effective decision-making in real situations. *Conclusions:* This integrated approach of mathematical modeling and discrete event simulation offers valuable insights to address disaster management and support decision-making in humanitarian crises.

**Keywords:** humanitarian logistics; disaster management; post-disaster relief logistics; redistribution; simulation discrete; logistics; optimization; mixed-integer programming; modeling

## 1. Introduction

Disasters caused by natural phenomena, such as earthquakes, are sudden and dangerous events that cause damage to human life. Since the early 2000s until now, more than 7350 disasters have been reported; these affected over 4 billion people and caused more than a million deaths and trillions in economic losses [1]. According to [2], the population affected by a natural disaster will increase in cities with a higher concentration of inhabitants; therefore, urban development requires a managing response and a decision-making plan in the face of a humanitarian crisis. On the other hand, some human actions can magnify the damage of natural disasters; for instance, the vulnerability of households increases the risk, understood as the expected damage from disasters, that they face.

Annually, cataclysms such as earthquakes, hurricanes, volcanic eruptions, and floods claim lives worldwide. Among the most devastating catastrophes of recent decades are the earthquake in Japan with more than 20,033 deaths [3], the Sichuan earthquake in China and the earthquakes in Haiti that affected more than 56 million people in both countries

and the Chilean earthquake of 2010 [4]. All these events caused significant damage to the population. According to recent statistics, Iran is the seventh country in the world with the highest risk of earthquakes [5,6]. Evidence shows earthquakes are the most frequent natural catastrophes [7]. Disaster management in an earthquake is one of the most critical issues. In addition to a large number of victims, the economy and health system of affected areas may deteriorate after disasters [8]. In this sense, places like Aceh (Indonesia) have substantially improved their disaster preparedness and mitigation over the years [9].

In Colombia, the consequences of nature's actions can also be evidenced; according to the study conducted [10], in the last 40 years, more than 28 thousand disaster events were recorded, of which 60% were reported from the 1990s onwards. During 2010 and 2019, a quarter of such records were reported, and a third of the population was affected. In this context, in 1983, Popayan registered an earthquake of 5.5 on the Richter scale; this situation meant a high humanitarian crisis [11]. There were more than 100,000 victims, 1200 injured, and 300 casualties; it is estimated that 2470 houses were destroyed and economic losses of 378 million dollars (0.98% of the G.D.P. of that year) [12]. This municipality is located in the Romeral fault area, which crosses the country from north to south in the Andean zone. This seismological fault originated from the Pacific Ring of Fire. In this case, there is sufficient historical evidence to highlight that if studies, actions, and plans that lead to a significant reduction in risk are not prioritized, disasters such as the one in 1983 could be repeated, which left a large part of the city in ruins [12].

Therefore, given the growing trend of disasters globally [13], it is relevant to keep studying issues related to humanitarian logistics. In addition, it is worth noting the particular requirements of a humanitarian supply chain compared to the traditional one: higher initial demand, the unpredictability of resources, and the relevance of decisions made [14]. The literature review shows diverse quantitative models that address different problems in humanitarian logistics. From the actors' perspective involved, the need to go beyond the benefits of theoretical models and incorporate organizational and human systems aspects that significantly affect disaster response is evident. This research seeks to integrate both paradigms by incorporating organizational and institutional aspects into quantitative and qualitative models that address operational problems.

In that sense, the objective of this article proposes a mixed methodology of mathematical modeling and discrete event simulation to analyze the impact of adopting coordination mechanisms related to information exchange and collaboration on the efficiency of disaster response when an emergency scenario and different sources of risk arise in the event of a possible earthquake in the city of Popayan. Although the solutions of both methodologies are different because mathematical modeling provides an exact solution while simulation modeling provides an approximate solution, the exact model is used as a basis in this research. Uncertainty or randomness elements are incorporated through simulation techniques to obtain a more realistic perspective of the System's behavior. Simulation techniques were also used to explore scenarios and evaluate how the exact model responds to different conditions and situations. This combined approach of exact and simulation models allowed for a more complete and robust approach to addressing the case study. Therefore, this research aims to answer the following question: How to integrate engineering techniques to coordinate actors in the supply of humanitarian aid for a possible earthquake in Popayan?

The remainder of this article is organized into four sections. Section 2 presents a summary of relevant literature. Section 3 describes the methodology used for the research. Section 4 analyses the results of the optimization and simulation model for each evaluated scenario. Finally, Section 5 presents the conclusions and discusses possible avenues for future research.

## 2. Literature Review

The literature on humanitarian logistics presents different approaches as reference development programs and policies under the integration of the public and private sectors [15],

their relationship with logistics and transport with a humanitarian approach [16,17], and the application of production planning in the business environment in crises [18]. In addition, the applied cases present the concepts of risk in the supply chain from a humanitarian perspective [19–21]. Some authors solved problems focused on the victims' needs [21], taking into account the cases of organizations' resilience and the emphasis on visibility and communication [22–24].

### 2.1. Humanitarian Logistics

Humanitarian Logistics (H.L.) is the process of efficiently planning, implementing, and controlling the cost-effective storage of goods and materials and related information from the point of origin to the point of consumption [15]. In addition, it is necessary to have the ability to recognize (detect), determine required resources (seizure), and review strategies to achieve effective supply chain operations in the face of disasters (reconfiguration/transformation) [25]. The H.L. application is essential to improve disaster management, and its development and challenges have become more critical in recent years due to increased activity worldwide [26]. Since 2001, about 500 events have occurred annually, with an average of about 75,000 casualties and more than 200 million people affected.

Humanitarian logistics (H.L.) for disaster response has been developed for several decades [27], aiming to improve efficiency and effectiveness in providing aid and assistance to communities affected by natural disasters and humanitarian emergencies. In this context, several studies have analyzed the role and performance of the tasks of each of the members of the organizations in charge of humanitarian aid in different catastrophic events, such as the hurricane that affected the Gulf Coast in the United States [27]. Over time, the application of methodologies and technology [28] has been promoted in research, seeking to optimize logistical processes and improve the capacity to respond to critical situations. However, despite these advances, humanitarian aid organizations still face challenges in the supply chain to reach the disaster-affected area promptly and efficiently and adequately meet the demand [29].

Moreover, the impact of H.L. in emergencies and the location of facilities or shelters are essential parts of the current research [30,31]. It ratifies the relevance of application for each type of calamity because, through this, the negative consequences caused by different disasters can be minimized.

### 2.2. Planning

It can be defined as managing and organizing different activities to minimize the impact of natural disasters on a specific population [6]. As mentioned, pre-positioning emergency supplies [32] is a mechanism to increase preparedness for natural disasters. It is a planning tool that determines the location and quantities of the different types of emergency supplies. As reported by [33], an essential component of H.L. is the inventory system since it can ensure an adequate supply of critical supplies to meet the needs of the victims.

Another relevant issue is evacuation planning; as mentioned by [34], in an emergency, evacuation is carried out to move people from a dangerous place to a safer one. Ref. [35] raised a robust method of scheduling and routing vehicles for emergency evacuation from wildfires with a study based in Australia. The theory of queuing networks was applied to the first work on disaster planning, as mentioned [36], in such a way that it was sought to design an optimal distribution network mitigating the inherent consequences of a disaster. The application of the multi-hierarchical criterion appeared hand in hand with routing models [14,37] developed a model that maximizes the coverage of the affected regions and minimizes human suffering through a social cost function. For example, some studies presented stochastic models in which the location of the site is chosen to satisfy the demand based on different possible catastrophes [32] or their impacts: Ref. [38] It also included budget restrictions before and after the disaster; the level of service quality [39]. Ref. [40] defined an optimal shelter network that minimizes transportation time while [40]

proposing a two-stage stochastic model to consider shelter provision. In [41] a multiperiod resource allocation optimization model was established in this study as an extension of the traditional model from the cross-regional collaborative perspective. Ref. [42] analyzed the operation strategies of specialized rescue teams in emergencies. The teams applied standard procedures for rescue operations. Stakeholders should align their values, purposes, and goals to their strategic planning process since this preparation occurs in an environment with a destabilized infrastructure and poorly competent transport connectivity [6].

### 2.3. Response

In 2006, a framework for applying indicators to H.L. processes [43] was created; it highlights the importance of using multimodal transport in response and recovery operations [44]. The heuristic algorithms applied a model in response operations that included routing, vehicle allocation, and merchandise distribution [38].

Natural disasters can cause severe casualties and economic losses, and emergency shelters are effective measures to reduce disaster risks and protect lives. Shelters have functions in different phases of disasters and can be defined as: "A place used for a short stay, ideally no more than a few weeks after the disaster" [45].

These models seek to choose shelters from a given set of alternate locations and provide transportation plans to minimize total costs or evacuation time [25]. They proposed a stochastic model for the distribution of aid and evacuation of victims towards the positioning of temporary care centers, intending to minimize the number of untreated injured and shortages of basic supplies. On the other hand, Ref. [40] used the Stackelberg game to guarantee locations by the communities, where the authorities act as leaders determining the locations of the shelters to minimize the total evacuation time. Ref. [46] formulated a multiobjective location model based on a goal programming approach that considers the uncertainties of damage to infrastructure due to earthquakes, Ref. [47] considered multiple risk objectives, number of sites, unsatisfied demand, and adequacy qualitative of locations. Along this line, Ref. [48] suggested a relief network that offers fixed and temporary hospitals, transfer points, ambulance stations, helicopter stations, and demand points.

Ref. [49] in addition, proposed a multi-pronged methodology using an optimization model and multi-criteria decision analysis to pre-position disaster relief supplies. Ref. [50] developed a dynamic model for dispatching and directing vehicles in response to an earthquake, focusing on transporting essential goods to affected areas and the injured to hospitals. Ref. [51] researched a novel and comprehensive algorithm to analyze an emergency location routing problem considering a fast and stable response to different situations. Ref. [51] proposed a multiobjective, multimodal, multi-period stochastic model for managing commodity and casualty logistics in earthquake response.

Ref. [52] addressed the location-allocation of medical services in emergency response with a coverage objective, forcing the satisfaction of a minimum demand [53]. For its part, this paper takes advantage of the experience with the COVID-19 pandemic to determine and control the dynamics of population mobility to reduce the impact by using an exposure indicator based on the movement ranges provided by Facebook to determine the dynamics of population mobility.

### 2.4. Recovery

It seeks to characterize disaster management at a general level by some international organizations, such as the United Nations Development and Environment Programmes (UNDP-UNEP), which identify four fundamental phases: prevention, preparation, response, and recovery [54]. As defined by many authors, they involve a series of planning and prevention processes, namely: natural risk assessment, prioritization of community objectives, tasks, and activities, identification of standards and indicators to measure efficiency and effectiveness, the establishment of coordination protocols between actors, inventory of local community capacities, the command center for management, means, measurement

and evaluation of results, corrective actions to improve the situation generated and alert system [55].

The quality and speed of logistics activities in the recovery phase impact how the community rebuilds from a disaster; therefore, these studies should be presented as a basis for long-term development programs and public policies, as suggested by [56]. In the context of post-disaster recovery, the quality and speed of logistics activities play a crucial role in the reconstruction of an affected community. In this sense, it is emphasized that the economy should be reactivated considering the principles of humanitarian logistics [57]. Specifically, assessing the demand for humanitarian aid is a key factor in planning recovery and effectively distributing resources.

## 3. Methodology

This type of research requires an initial treatment to discover and understand the empirical elements that hinder the implementation of coordination in the care of natural disasters [6]. The case study is an empirical research process where a contemporary phenomenon is analyzed in depth in its natural context [58], according to Figure 1.

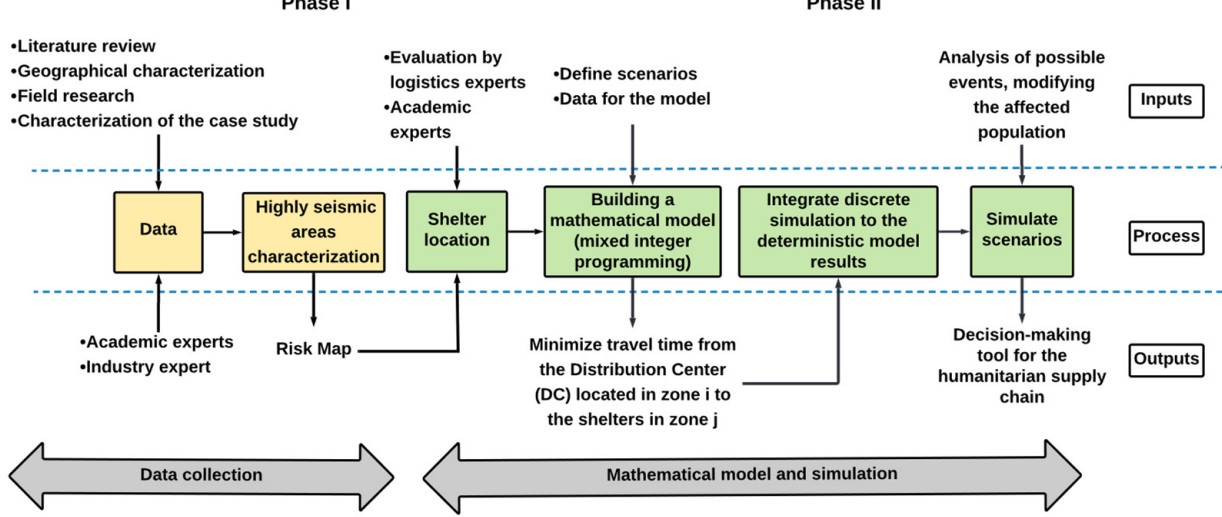

**Figure 1.** Methodological design.

The methodology proposed in this research provides the following elements:

- Definition of the problem: the bibliographic search and formulation of the problem were carried out with the antecedents of the Popayan earthquake in 1983.
- Deterministic model: consisted of determining the variables and parameters of the optimization model, of minimizing the times from the center's distribution of humanitarian aid to shelters in high seismic risk areas.
- Simulation model: integrate the results of the deterministic model through discrete simulation to combine a scenario analysis of possible events.
- Validation: validate the data of the deterministic models and simulation with the actors of the humanitarian logistics chain through seminars, workshops, and meetings.

### 3.1. Problem Definition

In case of an earthquake, Figure 2 shows the structure of the humanitarian chain considered in this investigation. The supplies are transferred from a supplier to the distribution centers. The supplier is the warehouse of Infantry Battalion No. 7 José Hilario López (Colombian National Army). For this study, we considered that, in the disaster aftermath, the relief supplies served all the population in need. The objective is to provide a model considering activities from the two phases. The optimal location for the distribution centers

of a set of potential sites is determined in the preparation phase. The closest point to which the affected people should move is determined in the post-disaster phase.

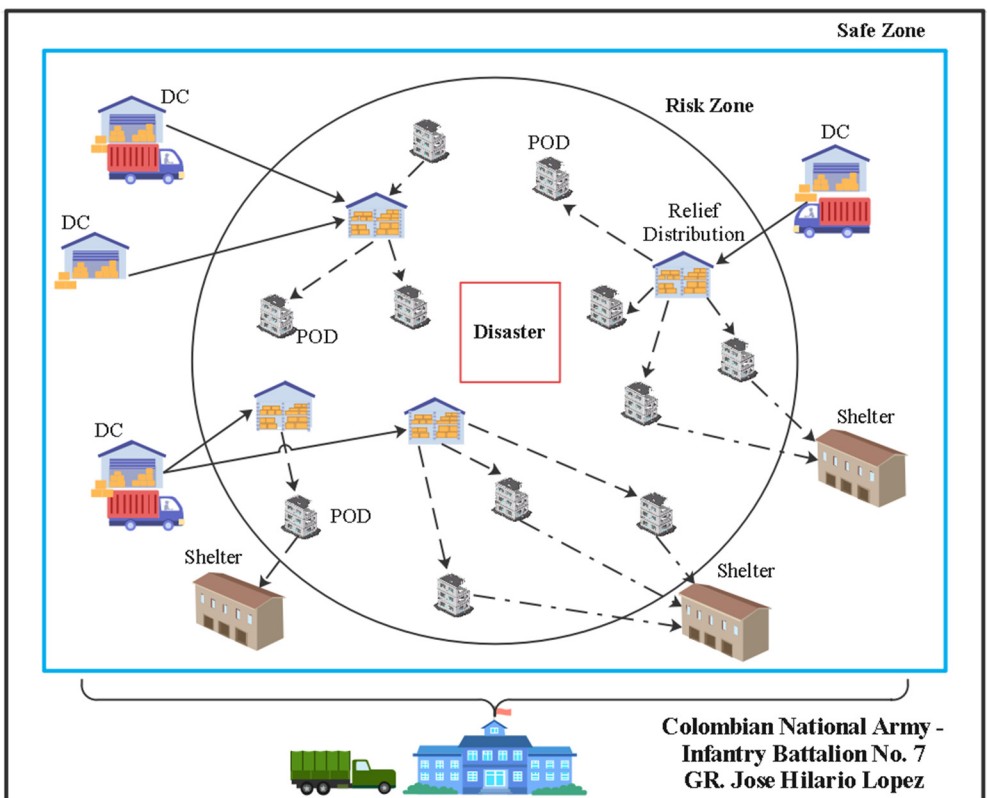

**Figure 2.** Diagram of the disaster areas. Source: Developed by the authors.

### 3.2. Deterministic Model

The Municipalidad de Popayan is located between the western and central mountain ranges in the southwest of Colombia, with an area of 464 km². According to the last population census in Colombia [59], the Municipalidad de Popayan has 318,059 inhabitants, with a population density of 587.57 inhabitants/km². According to the government of Popayan and the University of Cauca, in 2017, zoning was carried out to analyze the places with the highest risk and the population that would be affected by a seismic event, in which negative results were obtained, that is, the total of the inhabitants of the municipality are at high risk.

Table 1 presents the model's sets, parameters, and decision variables. Section 3.2.1 describes the objective function, while Section 3.2.2 details the structures of the constraints. Additionally, Section 3.2.3 distinguishes between binary and positive continuous variables.

#### 3.2.1. Objective Function

$$\text{Minz} \sum_{\forall j \in J, i \in I} t_{ij} \times U_{ij} + \sum_{\forall h \in H, j \in J} dz_{hj} \tag{1}$$

The objective function (1) has a clear purpose: to minimize the travel time from the Distribution Center (DC) located in zone *i* to the shelters in zone *j*. In addition, it seeks to reduce the distance between the affected areas and the nearest shelter. This strategic approach alleviates human suffering and optimizes allocating limited resources. By reducing response times and improving logistical efficiency, we can provide faster and more effective assistance to those who need it most.

**Table 1.** Sets of parameters and decision variables of the model.

| Index Sets | |
| --- | --- |
| I | potential location for the establishment of the distribution center ($i \in I$) |
| J | shelter's potential location ($j \in J$) |
| H | affected areas ($h \in H$) |
| **Parameters and Units** | |
| $t_{ij}$ | Travel time between the D.C. (Distribution Center) in zone $i$ and the shelter located in zone $j$ [min] |
| $dz_{hj}$ | Travel time from affected areas h to shelters in area $j$ [min] |
| $z_h$ | Number of families affected in zone h that require attention [people] |
| $c_j$ | Capacity of victims to attend shelters in zone $j$ [percentage] |
| $bigM$ | A very large arbitrary number of supplies |
| $cn$ | Consumption of supplies for each injured party [kits] |
| $S_i$ | DC distribution center capacity [kits] |
| **Decision Variables** | |
| $X_i$ | Binary variable. 1 to indicate if D.C. is located in zone $i$; 0 otherwise |
| $F_j$ | Binary variable. 1 to indicate if the shelter is located in zone $j$; 0 otherwise |
| $U_{ij}$ | Binary variable. 1 to indicate if D.C. $i$ serve shelter $j$; 0 otherwise |
| $XN_{hj}$ | Number of victims in zone k are cared for in shelter $j$ [people] |
| $Y_{hj}$ | Binary variable. 1 to indicate if the victims of zone h are cared for in shelter $j$; 0 otherwise |
| $Q_{ij}$ | Quantity of supplies to be sent from D.C. $i$, to shelter $j$ [units] |

### 3.2.2. Constraints

$$\sum_{\forall i \in I} X_i = 1 \tag{2}$$

Constraint (2) guarantees the opening of a single D.C. *i*.

$$\sum_{\forall i \in I} U_{ij} = F_j, \forall j \in J \tag{3}$$

Constraint (3) ensures that D.C. serves shelter *j* only if the shelter is open.

$$\sum_{\forall h \in H} XN_{hj} \leq \sum_{\forall h \in H} z_h \times F_j, \forall j \in J \tag{4}$$

Constraint (4) guarantees the care of the victims of each zone *h* in shelter *j* only if it is open.

$$XN_{hj} = Y_{hj} \times z_h, \forall j \in J, h \in H \tag{5}$$

Constraint (5) determines that a shelter serves the victims of the area only if it decides to serve them.

$$\sum_{\forall j \in J} Y_{hj} = 1, \forall h \in H \tag{6}$$

Constraint (6) guarantees that each victim in the affected areas *h* is cared for in a single shelter *j*.

$$\sum_{\forall h \in H} z_{hj} \geq 0.1 \times c_j \times F_j, \forall j \in J \tag{7}$$

Constraint (7) guarantees that a shelter opens only if the number of victims cared for exceeds 10% of the shelter's capacity. This value is the standard defined by the study area's relief agencies.

$$\sum_{\forall h \in H} z_h \times Y_{hj} \leq c_j \times F_j, \forall j \in J \tag{8}$$

Constraint (8) guarantees that each open shelter must have sufficient capacity to serve the victims of an affected h area.

$$Q_{ij} \leq U_{ij} \times bigM, \forall j \in J, j \in J \tag{9}$$

Constraint (9) ensures the shipment of supplies from a distribution center to the shelter only if the distribution center serves the shelter.

$$\sum_{\forall j \in J} Q_{ij} \leq s_i \times X_i, \forall i \in I \tag{10}$$

$$\sum_{\forall i \in I} Q_{ij} \leq cn \times c_j \times F_j, \forall j \in J \tag{11}$$

Constraints (10) and (11) ensure that the number of supplies sent from the distribution center to the shelters should not exceed the distribution center's capacity and the shelters' capacity, respectively.

$$\sum_{\forall j \in J} XN_{hj} \geq z_h, \forall h \in H \tag{12}$$

Constraint (12) guarantees that the total number of victims to be cared for in shelter j in zone h must be at least the total number of victims in that zone.

$$\sum_{\forall j \in J} F_j \geq \frac{\sum_{\forall h \in H} z_h}{\sum_{\forall j \in J} c_j} \tag{13}$$

Constraint (13) guarantees that shelters are sufficient to serve the affected population.

$$\sum_{\forall i \in I} Q_{ij} \geq cn \times \sum_{\forall h \in H} XN_{hj}, \forall j \in J \tag{14}$$

Constraint (14) ensures that sufficient supplies are shipped to meet the shelter's demand.

3.2.3. Definition of Variables

$$X_i, Fj, Uij, Yhj \in 0, 1, \forall i \in I, j \in J, h \in H \tag{15}$$

$$Y_{hj}, Qij \geq 0, \forall i \in I, j \in J, h \in H \tag{16}$$

*3.3. Simulation Model*

Simulation models are relatively common in formulating logistics management plans to consider the impact of decisions on managing a given system [60]. The main advantage of simulation in evaluating supply chain performance in a virtual environment is reducing the risk of costly mistakes before actual application [61]. This tool is an increasingly accepted research method for studying complex systems in humanitarian logistics [62].

Discrete event simulation is a tool to analyze the coordination mechanisms' impact on the disaster response efficiency in a possible earthquake in Popayan. The key distinction between simulation and mathematical modeling lies in the simulation's ability to introduce uncertainty or randomness, thus enriching the perspective and providing a more realistic behavior, conditions, and disaster situations. Consequently, it is possible to visualize possible effects, allowing humanitarian aid operations viability on different earthquake magnitudes and understanding the elasticity and sensitivity of humanitarian logistics to changes in parameters.

The research simulation model also identifies hospital capacities, calculates humanitarian aid distribution times, and maps viable transit routes. The intersection between the simulation and the mathematical model is essential for disaster response and evalu-

ating the humanitarian operations' robustness and logistics adaptability in this complex sociotechnical disaster.

Formulation of the Simulation Model

The results present various scenarios evaluated through simulation in the FlexSim software, whose methodological structure is shown in Figure 3. In stage 2, we defined which aspects of the real System would be represented in the model and which would not. In stage 3, the input data for the simulation model was identified, with the main basis being the results obtained by the mathematical model, i.e., the location of distribution centers and shelters.

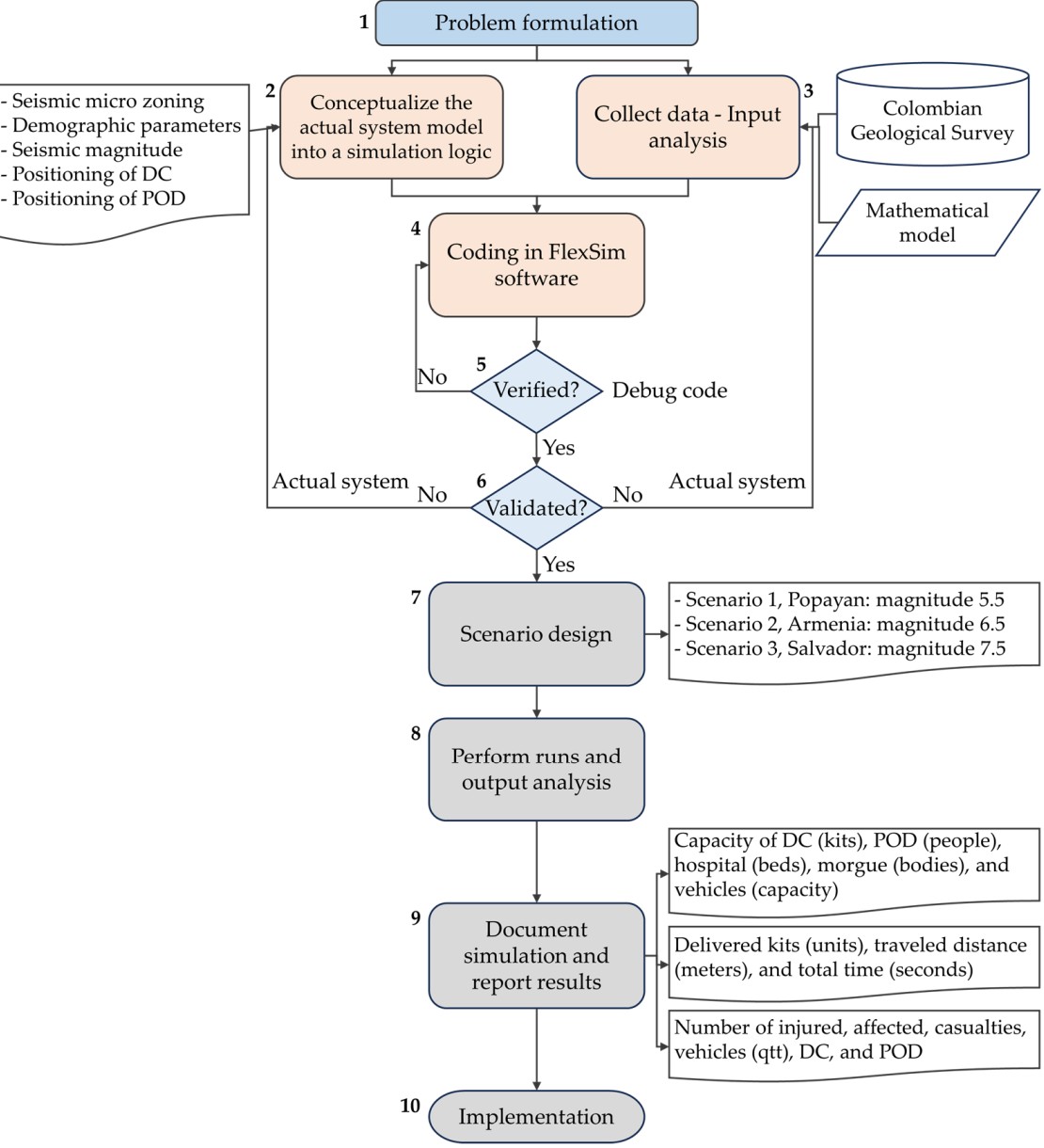

**Figure 3.** Simulation model formulation in Flexsim. Source: adapted from [63].

In stage 4, the simulation model was constructed based on the conceptual model of stage 2. In stage 5, the correct functioning of the simulation model was verified by analyzing the configuration of stage 4. In stage 6, it was validated that the simulation model is an exact representation of the real System, and the simulation model results were compared with those of the mathematical model.

For the analysis of the scenarios and the application of the simulation model, in stage 7, cases of earthquakes in Colombia and Latin America were proposed. In stage 8, the scenarios' executions and subsequent analyses were used to estimate the performance measures. Based on the analysis of the executions in step 9, it was determined if other executions were needed. In stage 10, the simulation model was documented to make decisions based on the analysis. Fourteen variants were determined and obtained.

## 4. Results

### 4.1. Definition of Distribution Centers and Points of Distribution

Threat areas identified in the risk maps provided by the Municipal Risk Unit (2020) were selected. They allowed four D.C. (Distribution Centers) and twenty possible P.O.D. (Point of Distribution). A humanitarian aid P.O.D. must cover the needs of each critical point, so the P.O.D. coverage area was defined as "at a distance of 500 m from the centroid" so that victims could access it [64].

Definition of Parameters and Variables

Humanitarian Aid in Emergencies (H.A.E.) should start immediately, scheduled for the first 48 h [65]., to maintain the level of food consumption of those affected as a result of a disaster. Each family registered and classified as affected must receive food assistance in the first hours established after the event [66]. Due to its composition, it will have an average of 8 to 10 days in a standard family nucleus of no more than five members (U.N.G.R.D., 2013). The characteristics of the four D.C.s (see Table 2) that were identified in the model are the following: an area of influence of 7.83 km$^2$, a capacity of 10 tons (capKg in kilograms), a fixed annual cost (fixed cost of facilities in U.S. dollars) and an area of 200 m$^2$.

**Table 2.** Humanitarian aid georeference distribution centers (D.C.).

| DC | DC Abbreviation | Capacity | Latitude | Longitude |
|---|---|---|---|---|
| Unidad de gestión de riesgos | UNGDR | 720 | 2.45260759 | −76.590884 |
| Complejo Deportivo Juegos Nacionales | CDJN | 1440 | 2.48721606 | −76.586081 |
| Coliseo La Estancia | CLE | 500 | 2.45121419 | −76.59807 |
| Polideportivo CDU | CDU | 2000 | 2.44754985 | −76.598616 |

The 20 geo-positioned P.D.O.s are in Table 3; column 1 shows the coding of the distribution point, column 2 shows the current use or category, column 3 indicates the name, and finally, the last two columns show the latitude and longitude. These are public places with enough space to attend to the victims.

### 4.2. Deterministic Model Results

The solution for the Popayan case study model was carried out in the C.P.L.E.X. solver and GAMS modeler using structured language; there were four executions with 2500 iterations each, 10,000 iterations in total. Each iteration began with the random sampling of the variables and continued with the solution in stages.

The mathematical model determined that 3 H.A.E. distribution C.D.O.s should be activated: Complejo Deportivo (C.D.J.N.), Coliseo La Estancia (C.L.E.), and Polideportivo. In addition, 4 P.O.D.s were stipulated for the care of the victims: Comfacauca Olympic Village (V.O.C.), La Estancia Coliseum (C.L.E.), CDU Sports Center (CDU), and Tablazo (T.B.). Neighborhoods served in each shelter: Antonio Nariño (AN), Santa Clara (S.C.), Modelo (M), Nueva Granada (N.G.), La Cabaña (L.C.), La Playa (L.P.), Plazuela del Poblado (P.P.), Block of Pubenza (B.P.), Prados del Norte (P.N.), Belalcázar (B.E.), El Recuerdo (E.R.) and Loma Linda (L.L.). Table 4 summarises the distribution centers that supply the affected areas' service points with the number of families and kits served.

**Table 3.** Distribution points (P.O.D.) georeferences.

| N | Zone | POD | POD | Capacity | Latitude | Longitude |
|---|------|-----|-----|----------|----------|-----------|
| 1 | North Zone | Polideportivo Villa del Viento | PVV | 100 | 2.481 | −76.578 |
| 2 | North Zone | Polideportivo Las Guacas | PLG | 100 | 2.474 | −76.550 |
| 3 | North Zone | Coliseo La Estancia | CLE | 100 | 2.451 | −76.598 |
| 4 | North Zone | Villa Olímpica Comfacauca | VOC | 200 | 2.460 | −76.599 |
| 5 | Downtown Zone | Polideportivo Pandiguando | PG | 100 | 2.448 | −76.616 |
| 6 | North Zone | Polideportivo Palace | PP | 100 | 2.459 | −76.590 |
| 7 | North Zone | Polideportivo San Camilo | PSC | 100 | 2.437 | −76.610 |
| 8 | Eastern Zone | Parque El Recuerdo | PER | 50 | 2.452 | −76.599 |
| 9 | South Zone | Polideportivo El Lago | PLG | 181 | 2.432 | −76.597 |
| 10 | Downtown Zone | Polideportivo Sindical 1 | PS1 | 100 | 2.431 | −76.611 |
| 11 | North Zone | Polideportivo Retiro Alto | PRA | 100 | 2.441 | −76.624 |
| 12 | North Zone | Polideportivo Retiro Bajo | PRB | 100 | 2.441 | −76.621 |
| 13 | North Zone | Polideportivo Colombia | PC | 100 | 2.451 | −76.634 |
| 14 | North Zone | Polideportivo María Occidente | PMO | 100 | 2.455 | −76.632 |
| 15 | Downtown Zone | Polideportivo Colombia 2 | PC2 | 100 | 2.454 | −76.632 |
| 16 | Downtown Zone | Polideportivo El Mirador | PEM | 100 | 2.445 | −76.628 |
| 17 | Downtown Zone | Polideportivo CDU | CDU | 200 | 2.447 | −76.598 |
| 18 | North Zone | Centro Recreativo Pisoje | CRP | 200 | 2.477 | −76.571 |
| 19 | Downtown Zone | Centro Recreativo Guayacanes | CRG | 200 | 2.459 | −76.639 |
| 20 | North Zone | Tablazo | TB | 1000 | 2.473 | −76.582 |

**Table 4.** Mathematical model results.

| Distribution Center DCO | POD Attended by D.C.O. | Victims of the Area Treated in the P.O.D. | Affected Families Cared for in the P.O.D. | Kits Sent to P.O.D. | % P.O.D. Attention |
|---|---|---|---|---|---|
| BOM 1 | INEM 5 | AN1-SC12 | 192 | 384 | 96% |
| | ISF 9 | M8-NG9 | 126 | 252 | 63% |
| UNGDR 2 | CSM 16 | LC5-LP6-PP10 | 168 | 336 | 93% |
| CRC 3 | INJP 19 | BP3-PN11 | 167 | 334 | 93% |
| DVC 4 | C.C.H. 6 | BE2 | 163 | 326 | 92% |
| | HSM2 | ER4-LL7 | 184 | 368 | 96% |
| Total | | | 1000 | 2000 | |

In a disaster, the humanitarian logistics chain must react immediately and provide the best assistance to affected people (see Figure 4 for the location of distribution centers). The model proposed in this article is designed to serve 1000 families from 12 neighborhoods in communes 1 and 2 of Popayan, with a response time of 92 min to deliver 2000 H.A.E. kits.

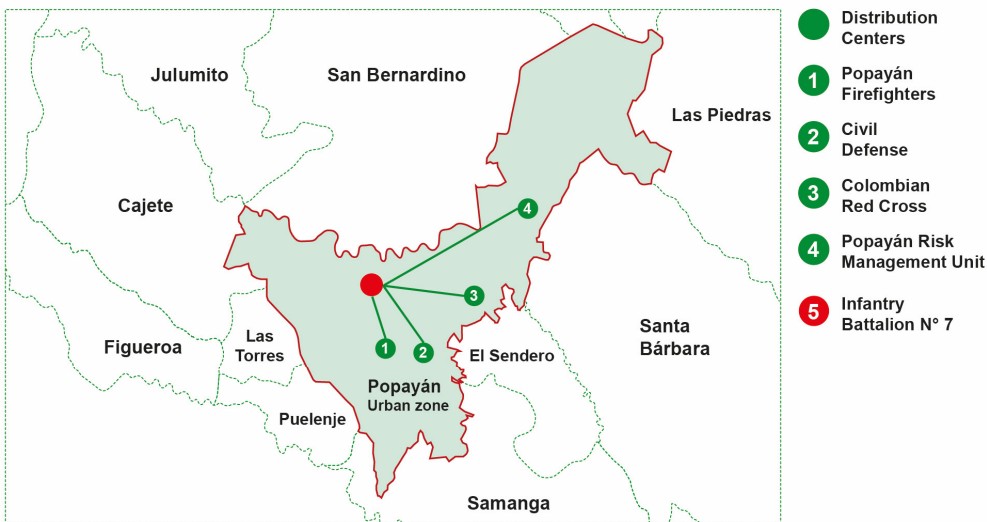

**Figure 4.** Humanitarian aid distribution centres. Source: Google Maps, 2021.

*4.3. Simulation Model Results*

The results from the mathematical model presented in Table 4 were used as the basis for the simulation model, and the following scenarios were evaluated.

Simulated Scenarios

Real cases were proposed in Colombia and Latin America to analyze the scenarios. The first scenario refers to the 1983 Popayan earthquake, with a magnitude of 5.5 on the Richter scale. Humanitarian aid focused on Popayan, where most deaths were reported, with approximately 300 casualties, 1500 injured, and 13,796 homes severely damaged.

The second scenario represents the most catastrophic earthquake in Colombia that occurred on 25 January 1999, leaving a balance of 1125 casualties and 2300 injured. More than 30,000 homes were affected with a magnitude of 6.5 on the Richter scale, and about 75% of the Armenian city was destroyed, generating quantified damage equivalent to 2.2 of G.D.P. in 1998.

The last scenario is one of the most destructive earthquakes in Latin America, specifically in El Salvador, which occurred on Friday, 10 October 1986, with a magnitude of 7.5 on the Richter scale, leaving around 5600 casualties, 200,000 affected and 20,000 injured [65].

Simulation experiments were carried out. Fourteen variants were adopted to evaluate the sensitivity of the solution obtained. These variants are defined as follows: (1) D.C. capacity (kits), (2) P.O.D. capacity (people), (3) hospital capacity (beds), (4) morgue capacity (bodies), (5) capacity of vehicles, (6) kit delivered (units), (7) distance travelled (meters), (8) total time (seconds), (9) the number of injured, (10) the number of affected, (11) the number of casualties, (12) the number of vehicles, (13) the number of D.C. and (14) the number of P.O.D.

*4.4. Simulated Results (Experiments)*

Simulations were carried out for 14 defined variants. Each variant had 30 repetitions, mainly for variants 9, 10, and 11, in different random samples for the respective distributions. Figure 5 identifies the number of vehicles used, D.C., and P.O.D. in each simulated scenario.

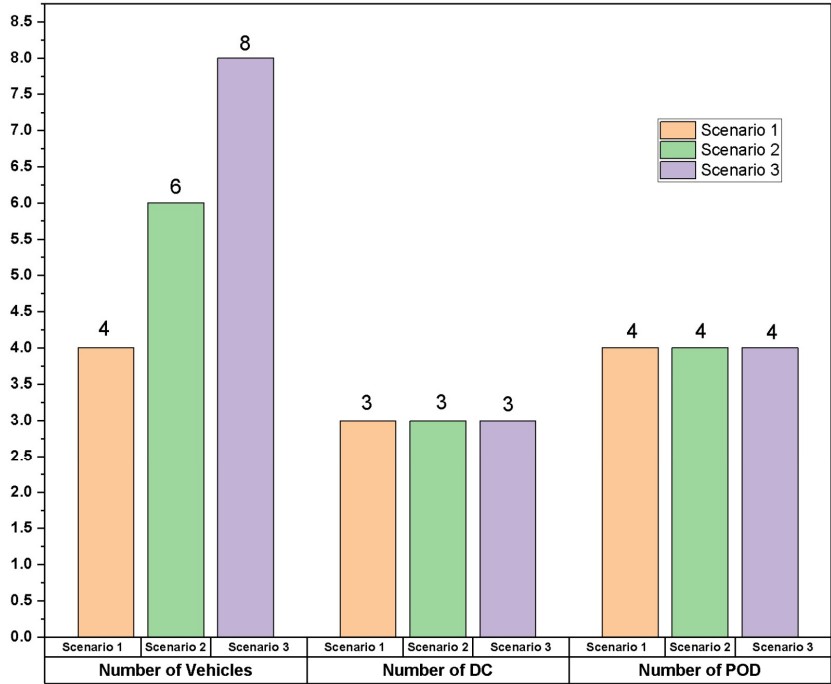

**Figure 5.** Resources used in the simulation scenarios. Source: Developed by the authors.

This article focuses on evaluating the sensitivity of the solution, which means an evaluation of the viability of humanitarian aid operations in the event of an earthquake at different magnitudes on the Richter scale. Figure 6 presents the number of affected, injured, and casualties.

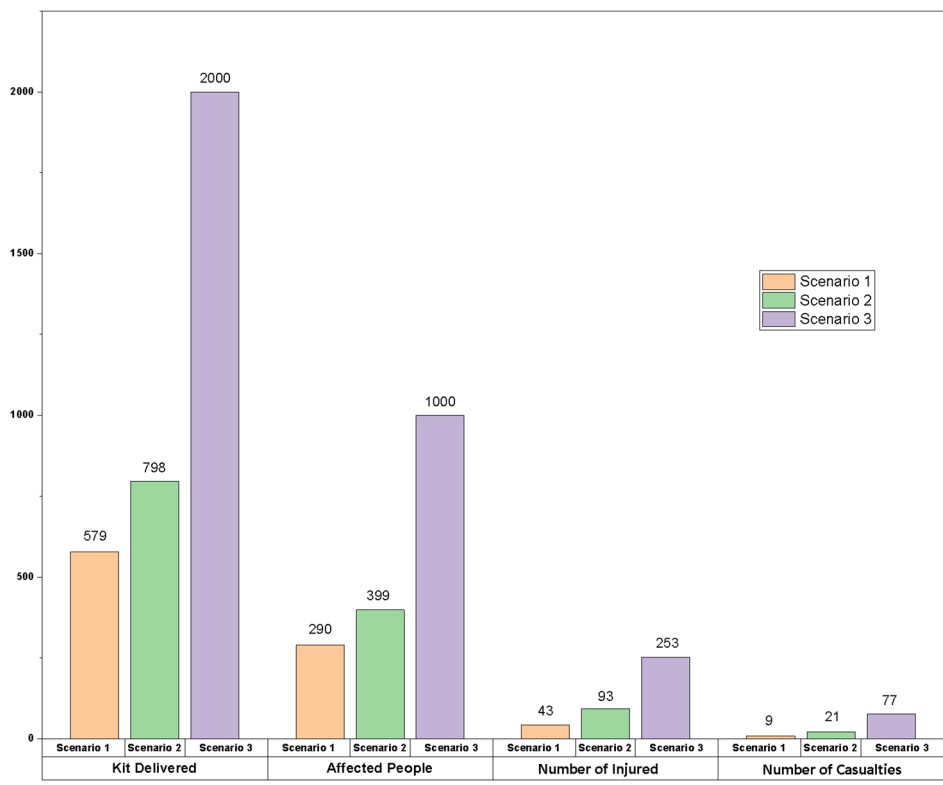

**Figure 6.** The number of people affected, injured, and casualties. Source: Developed by the authors.

Another characteristic was the evaluation of the utilization of the installation's capacity. Figure 7 shows the occupancy values for distribution centers, shelters, hospitals, and morgues.

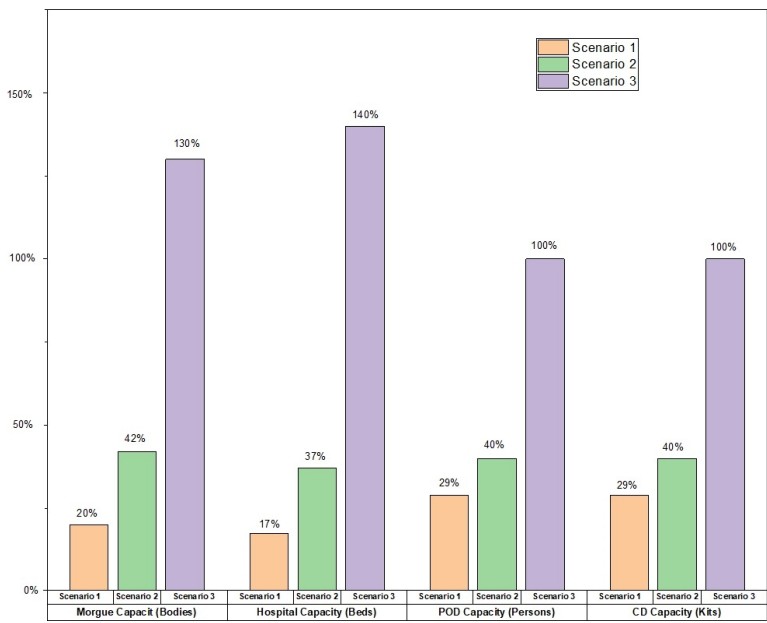

**Figure 7.** Percentage of Occupation of hospitals, morgue, D.C., and P.O.D. Source: Developed by the authors.

This research is limited to examining the previous characteristics because they were considered representative to evaluate the elasticity and sensitivity in humanitarian logistics to parameter changes. This model allows evaluation of the goodness of the solution returned for the formulated decision problem. As shown in Figure 8, the experiments, deterministic variants with the same intensity of affectation, allow the implementation of the aid plan in 2.4 h and travel a distance of 21.28 km for the most critical scenario evaluated.

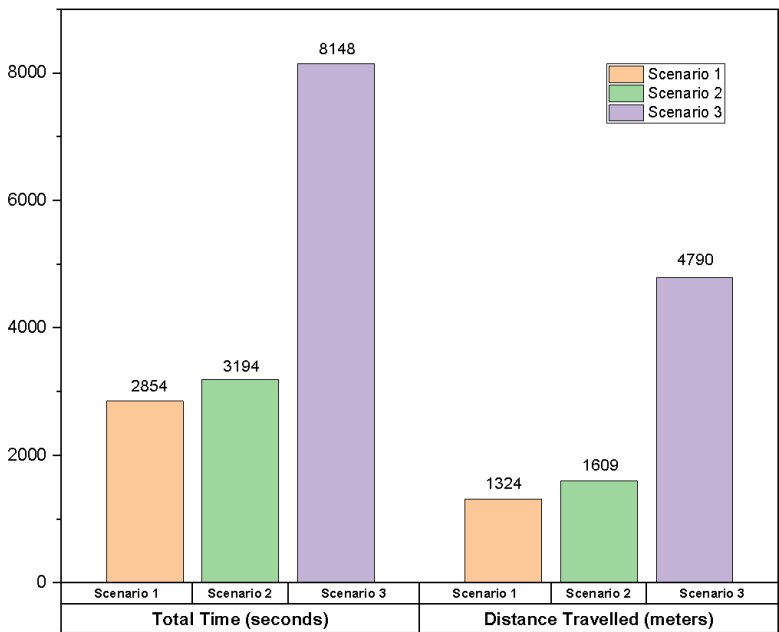

**Figure 8.** Implementation time and route for humanitarian aid. Source: Developed by the authors.

## 5. Validation

To validate the feasibility and coherence of the results, they were presented to the Disaster Risk Management Committee of the city of Popayan, which are experts responsible [67] for helping to prevent and mitigate the impact caused by natural risks. This committee stated that the results are a useful tool to support decision-making regarding the location of distribution centers and shelters.

In addition, the first Humanitarian Logistics Seminar was held at the Risk Management Unit (U.N.D.G.R.), where experiences were shared about the Popayan and Eje Cafetero earthquakes. The importance of standardization in care was highlighted to ensure that victims receive reasonable quality assistance through a participatory system expressing community resilience. The existing regulations for humanitarian aid, such as Law 1523 of 2012, were emphasized.

On the other hand, a seismic microzones workshop was carried out with the Firefighters of the city of Popayan and the U.N.D.G.R. to characterize the high seismic risk zones according to the Romeral faults located between the central and western cordilleras, as well as the cracks caused by the 1983 earthquake.

It is important to highlight that for the empirical application of this model, there must be a consensus among the involved actors. This study is expected to be implemented in the methodologies currently structured by government entities dedicated to natural disaster response.

## 6. Conclusions

This study generated many expectations on managing earthquakes and eventualities, since in the 1983 earthquake alone, more than 100,000 people were affected in Popayan. For this reason, it is very satisfactory to generate contributions on the appropriate management and concrete decision-making in the entities in charge of organizing and directing the aid processes to those affected by a disaster, achieving a response time of 79 min for the delivery of 2000 humanitarian aid kits. This model presents how the humanitarian logistics network strengthened the knowledge of the institutions to transfer a more practical tool for managing the logistics model that contributes to maintaining the safety of the inhabitants.

After the Popayan earthquake in 1983, construction methods were developed taking into account specific soil studies and architectural needs that depend on the programs that arise from each guideline, such as the P.O.T. (Plan de Ordenamiento Territorial), where each of the parameters required in urban and rural areas must be fully complied with to improve construction projects, thus avoiding infrastructure risks that affect human integrity. To date, each of the actors of the humanitarian chain is strengthened since the actors are known and determined, strengthening the different activities in the face of disasters caused by natural phenomena.

With this research analysis, it was possible to identify possible scenarios in the event of an earthquake, establishing four zones in Popayan and 20 possible points of attention with their respective humanitarian aid distribution centers and their reference capacities, which were not detailed before the model. The model clearly defines the logistical network in which the capacity of each of the shelters is available, which arises thanks to the research carried out.

One main limitation of this study is that it focuses on the Popayan downtown area, and the supplies can serve all the needy population. According to relief agencies, this is a high-risk area. Consequently, the has been designed to serve 1000 families from 12 neighborhoods in communes 1 and 2 of the city. Likewise, it is highlighted that the scenarios planned derive from past events that impacted central areas. The 1983 Popayan earthquake is characterized by a magnitude of 5.5 on the Richter scale. On this occasion, humanitarian assistance was concentrated downtown, with 300 casualties, 1500 people injured, and 13,796 homes severely damaged. Secondly, the study contemplates the most catastrophic earthquake in Colombia on 25 January 1999, which left 1125 casualties and 2300 injured. More than 30,000 houses were affected due to a magnitude of 6.5 in the

Richter escalation, causing considerable devastation in Armenia, where about 75% of the city was reduced to rubble, resulting in economic damage equivalent to 2.2% of G.D.P. in 1998. Finally, the third scenario is linked to one of the most devastating earthquakes in Latin America, which occurred in El Salvador on 10 October 1986. Characterized by a magnitude of 7.5 on the Richter scale, this earthquake left around 5600 victims, 200,000 affected, and 20,000 injured.

**Author Contributions:** Conceptualization, H.P.-O., J.A.S.D., M.C. and I.d.B.J.; methodology, J.A.S.D., M.C., M.M., Y.A.-M. and I.d.B.J.; software, J.A.S.D., H.P.-O. and M.M.; validation, M.C., I.d.B.J. and Y.A.-M.; formal analysis, H.P.-O., J.A.S.D. and I.d.B.J.; research, H.P.-O., J.A.S.D., M.C., M.M. and Y.A.-M.; writing—preparation of the original draft, H.P.-O., J.A.S.D. and M.C.; writing—revision and editing, M.C., I.d.B.J. and Y.A.-M.; visualization, H.P.-O.; supervision, H.P.-O. and M.C.; obtaining funding, I.d.B.J. All authors have read and agreed to the published version of the manuscript.

**Funding:** Helmer Paz, Jhon Alexander Segura, and Yesid Ediver Anacona Mopan gratefully acknowledge the support of the Corporación Universitaria Comfacauca-Unicomfacauca. Irineu de Brito Junior is additionally grateful to National Council for Scientific and Technological (CNPq), 404803/2021-0 and Coordination for the Improvement of Higher Education Personnel-Brazil (CAPES), Procad Defesa 8887.387760/2019-00, Brazil. Mario Chong and Mariana Moyano would like to acknowledge the Universidad del Pacífico Research Center's (CIUP) contribution to developing this paper by providing crucial financial support under its grant.

**Institutional Review Board Statement:** Not applicable.

**Informed Consent Statement:** Not applicable.

**Data Availability Statement:** http://bit.ly/3zo5rA1 accessed on 31 August 2023.

**Conflicts of Interest:** The authors declare no conflict of interest.

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
