# Peer review of "Earthquake Decision-Making Tool for Humanitarian Logistics Network: An Application in Popayan, Colombia"

_logistics, 2023_

Round 1

Reviewer 1 Report

Dear Editor, 

Thank you for the opportunity to review the paper entailed: "Earthquake decision-making tool for humanitarian logistics network. Applied in Popayan, Colombia", submitted to Logistics

The authors propose an operational model for distributing and redistributing relief items in a post-disaster situation based on a mathematical programming model and simulation. 

·        The abstract is written as a technical report. An abstract must tell the merit of the paper and its unique contribution. I am asking the authors to modify the abstract according to this comment.

·        There are many references, but many of them are too old. For example, in the literature review, the authors survey the issue of pre-positioning emergency supplies by references [33] and [34], which are more than ten years old, while there are many newer papers. One of them for example, is:

Keren, B. Prepositioning Emergency Inventories Under Uncertainties of Time, Location, and Quantity. Oper. Res. Forum 4, 26 (2023). https://doi.org/10.1007/s43069-023-00211-w

Another very relevant missing paper for post-disaster is:

Sakiani, R., Seifi, A., & Khorshiddoust, R. R. (2020). Inventory routing and dynamic redistribution of relief goods in post-disaster operations. Computers & Industrial Engineering140, 106219.

My recommendation to the authors is to update their literature review and the references to the "state of the art", and to emphasize their contribution compare to the most recent papers.

·        As I understand, [50] were who proposed the relief network shown in Figure 1. If so, figure 1 belongs to another journal and can't be republished here without permission of the journal. A similar comment is about Figure 2, which was adapted from [12]. In Figure 3, the right-up distribution center should be denoted by DC (and not CD).

·        The authors present "Linear programming" as a keyword, but I do not see any linear programming model in the paper. If the deterministic model in section 3.2 is an "integer linear programming model", the authors should state that explicitly. Another keyword is "Vehicle routing problem". However, I did find in the model how this problem was applied or integrated into the model.

·        It is difficult to read the paper because it has too many abbreviations. See for example, the abbreviations of locations in lines 365-372. The authors must work on a more reader-friendly version. I am not sure that the accurate names and the locations of each DC and POD help to understand the model, and maybe a network flows diagram can be more informative.

·        The paper's core is the deterministic model, with the objective function (3.2.1) and the constraints (3.2.2). An objective function that minimizes the sum traveling time does not make sense in the problem contest. Some of the traveling can be done in parallel with several vehicles, so the time for ending the distribution of all the items is not a simple sum of the traveling time. However, I don't think that even minimizing the distribution timing of the last item is a good objective function. I believe that a disaster management plan should integrate cost, time, and operation considerations under limited resources. 

·        I also have some problems with the constraints. Constraint (14) ensures that sufficient supplies are shipped to meet the shelter's demand. However, what if a shortage happens and the demand is more than available? Constraint (14) prevents any solution, but in practice, it is clear that only part of the demands will be supplied. A typical situation in disaster management is a shortage of supply, so the authors must modify their model. Moreover, Constraint (7) guarantees that a shelter opens only if the number of victims cared for exceeds 10% of the shelter's capacity. This is an arbitrary economic constraint that is not related to the current objective function. The decision to open a shelter can be modeled by adding some fixed cost to any opened shelter, adding this somehow to the objective function, and letting the optimization process to decide if it is worth opening or not. 

The idea of distributing and redistributing relief items in a post-disaster situation, based on mathematical programming, is not a new concept but can be interesting, especially with a real case study. However, in my opinion, the paper is not ready for publication, further work is needed, and my recommendation is a "Major Revision". 

Dear Editor, 

Thank you for the opportunity to review the paper entailed: "Earthquake decision-making tool for humanitarian logistics network. Applied in Popayan, Colombia", submitted to Logistics

The authors propose an operational model for distributing and redistributing relief items in a post-disaster situation based on a mathematical programming model and simulation. 

·        The abstract is written as a technical report. An abstract must tell the merit of the paper and its unique contribution. I am asking the authors to modify the abstract according to this comment.

·        There are many references but many of them are too old. For example, in the literature review, the authors survey the issue of pre-positioning emergency supplies by references [33] and [34], which are more than ten years old, while there are many newer papers. One of them for example is:

Keren, B. Prepositioning Emergency Inventories Under Uncertainties of Time, Location, and Quantity. Oper. Res. Forum 4, 26 (2023). https://doi.org/10.1007/s43069-023-00211-w

Another very relevant missing paper for post-disaster is:

Sakiani, R., Seifi, A., & Khorshiddoust, R. R. (2020). Inventory routing and dynamic redistribution of relief goods in post-disaster operations. Computers & Industrial Engineering140, 106219.

My recommendation to the authors is to update their literature review and the references to the "state of the art", and to emphasize their contribution compare to the most recent papers.

·        As I understand, [50] were who proposed the relief network shown in Figure 1. If so, figure 1 belongs to another journal and can't be republished here without permission of the journal. A similar comment is about Figure 2, which was adapted from [12]. In Figure 3, the right-up distribution center should be denoted by DC (and not CD).

·        The authors present "Linear programming" as a keyword, but I do not see any linear programming model in the paper. If the deterministic model in section 3.2 is an "integer linear programming model", the authors should state that explicitly. Another keyword is "Vehicle routing problem". However, I did find in the model how this problem was applied or integrated into the model.

·        It is difficult to read the paper because it has too many abbreviations. See for example, the abbreviations of locations in lines 365-372. The authors must work on a more reader-friendly version. I am not sure that the accurate names and the locations of each DC and POD help to understand the model, and maybe a network flows diagram can be more informative.

·        The paper's core is the deterministic model, with the objective function (3.2.1) and the constraints (3.2.2). An objective function that minimizes the sum traveling time does not make sense in the problem contest. Some of the traveling can be done in parallel with several vehicles, so the time for ending the distribution of all the items is not a simple sum of the traveling time. However, I don't think that even minimizing the distribution timing of the last item is a good objective function. I believe that a disaster management plan should integrate cost, time, and operation considerations under limited resources. 

·        I also have some problems with the constraints. Constraint (14) ensures that sufficient supplies are shipped to meet the shelter's demand. However, what if a shortage happens and the demand is more than available? Constraint (14) prevents any solution, but in practice, it is clear that only part of the demands will be supplied. A typical situation in disaster management is a shortage of supply, so the authors must modify their model. Moreover, Constraint (7) guarantees that a shelter opens only if the number of victims cared for exceeds 10% of the shelter's capacity. This is an arbitrary economic constraint that is not related to the current objective function. The decision to open a shelter can be modeled by adding some fixed cost to any opened shelter, adding this somehow to the objective function, and letting the optimization process to decide if it is worth opening or not. 

The idea of distributing and redistributing relief items in a post-disaster situation, based on mathematical programming, is not a new concept but can be interesting, especially with a real case study. However, in my opinion, the paper is not ready for publication, further work is needed, and my recommendation is a "Major Revision". 

Author Response

Reviewer # 1’s Comments

Answer to Reviewer

Implemented changes

 The abstract is written as a technical report. An abstract must tell the merit of the paper and its unique contribution. I am asking the authors to modify the abstract according to this comment.

Thank you for your comment. We have modified the summary according to your suggestion and have highlighted both the unique contribution and the merit of our work in it. We hope it now meets your expectations.

Abstract Restructuring

There are many references but many of them are too old. For example, in the literature review, the authors survey the issue of pre-positioning emergency supplies by references [33] and [34], which are more than ten years old, while there are many newer papers.

New and more up-to-date references have been incorporated to complement the information. Some old references were changed (for example [16]) and we keep others just for highlighting basic concepts of humanitarian logistics.

Referencing of new citations

As I understand, [50] were who proposed the relief network shown in Figure 1. If so, figure 1 belongs to another journal and can't be republished here without permission of the journal. A similar comment is about Figure 2, which was adapted from [12]. In Figure 3, the right-up distribution center should be denoted by DC (and not CD).

Figure 1 was eliminated by decision of the work team.

Figure 2 is not identical to the one shown in the cited article. Instead, it was taken as inspiration to create the figure.

In Figure 3, the abbreviation CD has been changed to DC.

Figure 1 was eliminated.

The abbreviation CD has been changed to DC, where a new figure is created.

The authors present "Linear programming" as a keyword, but I do not see any linear programming model in the paper. If the deterministic model in section 3.2 is an "integer linear programming model", the authors should state that explicitly. Another keyword is "Vehicle routing problem". However, I did find in the model how this problem was applied or integrated into the model.

Thank you for your comment. The words "linear programming" and "integer linear programming model" are changed to "mixed integer programming", which is more in line with the objective of the proposed study.

“The keyword ‘Vehicle routing problem’ has been changed to ‘Disaster management’ in our study. This change was made because the problem addressed in our research is not fully linked to the concept of ‘Vehicle routing problem’. Instead, it is more closely related to disaster management.”

The words "linear programming" and "integer linear programming model" are changed to "mixed integer programming".

The keyword “Vehicle routing problem” has been changed to “Disaster management” in our study.

It is difficult to read the paper because it has too many abbreviations. See for example, the abbreviations of locations in lines 365-372. The authors must work on a more reader-friendly version. I am not sure that the accurate names and the locations of each DC and POD help to understand the model, and maybe a network flows diagram can be more informative.

We reviewed all the paper and the abbreviations presented in this article are explained in detail. For example, DC (Distribution Centers) on line 352, likewise P.O.D (Point of Distribution) is on line 353. In addition, some abbreviations are defined in Table 2, which minimizes the repetition of words.

The paper's core is the deterministic model, with the objective function (3.2.1) and the constraints (3.2.2). An objective function that minimizes the sum traveling time does not make sense in the problem contest. Some of the traveling can be done in parallel with several vehicles, so the time for ending the distribution of all the items is not a simple sum of the traveling time. However, I don't think that even minimizing the distribution timing of the last item is a good objective function. I believe that a disaster management plan should integrate cost, time, and operation considerations under limited resources.

Thank you for your comment. We explained the objective function and used references to support it. The objective function you mentioned was designed to reduce the suffering of people (according to Holguin Veras et al), so it makes sense to reduce time. The restrictions, on the other hand, were established to reduce the deprivation cost with a more efficient distribution of resources. By balancing these two objectives, a more effective and efficient disaster management plan can be achieved.

Lines 265-271

 I also have some problems with the constraints. Constraint (14) ensures that sufficient supplies are shipped to meet the shelter's demand. However, what if a shortage happens and the demand is more than available? Constraint (14) prevents any solution, but in practice, it is clear that only part of the demands will be supplied. A typical situation in disaster management is a shortage of supply, so the authors must modify their model. Moreover, Constraint (7) guarantees that a shelter opens only if the number of victims cared for exceeds 10% of the shelter's capacity. This is an arbitrary economic constraint that is not related to the current objective function. The decision to open a shelter can be modeled by adding some fixed cost to any opened shelter, adding this somehow to the objective function, and letting the optimization process to decide if it is worth opening or not.

Sufficient supplies is an assumption of the model. We clarified it in the problem definition.

Constraint (7) The reason for selecting the value of 10% is that this value has been given by the relief agencies in the study area and has been validated by them. We modified the constraint description to clarify it.

Lines 234-236

Lines 296-298

Reviewer 2 Report

The ideas in this paper are good. However, this article still need to be revised as follows:

1.     The Introduction and Literatures review need to be enhanced, especially on routing optimization problem, humanitarian Logistics and solution algorithms, and articles in recent years. These articles may be helpful for improving this paper: a hybrid metaheuristic algorithm for the multi-objective location-routing problem in the early post-disaster stage, the fourth-party logistics routing problem using ant colony system-improved grey wolf optimization, 4pl routing problem using hybrid beetle swarm optimization.

2.     The emergency decision-making scenario studied in this paper is under seismic conditions, so the relevant seismic information should be explained in the second paragraph of the Introduction, and the introduction of tsunami information is not easy to highlight the key points.

3.     The meaning of the red and green five-pointed stars in the relief network of fixed and temporary hospitals in Fig. 1 is unclear, and the relief routes are not easy to distinguish between fixed and temporary hospitals.

4.     Whether the objective function (formula (1)) is written in a standard way?

5.     The positions of Table 1 should be reformatted and should not be inserted between constraints.

6.     Formula (7) mentions that the shelter will only be opened when the number of victims being cared for exceeds 10% of the shelter's capacity. What is the reason for selecting the 10% value?

Minor editing of English language required

Author Response

Reviewer # 2’s Comments

Answer to Reviewer

Implemented changes

The Introduction and Literatures review need to be enhanced, especially on routing optimization problem, humanitarian Logistics and solution algorithms, and articles in recent years. These articles may be helpful for improving this paper: a hybrid metaheuristic algorithm for the multi-objective location-routing problem in the early post-disaster stage, the fourth-party logistics routing problem using ant colony system-improved grey wolf optimization, 4pl routing problem using hybrid beetle swarm optimization.

We clarified that the objective of the model is to reduce suffering. In the paper, a simple model is proposed so that decision-makers can interpret the results.

Introduction and Literature review was reviewed to clarify it.

The emergency decision-making scenario studied in this paper is under seismic conditions, so the relevant seismic information should be explained in the second paragraph of the Introduction, and the introduction of tsunami information is not easy to highlight the key points.

We agree to give greater importance to contextualizing the concept of earthquakes. Therefore, we have updated the second paragraph of the introduction following the recommendation.

The second paragraph was modified as suggested by the evaluator.

The meaning of the red and green five-pointed stars in the relief network of fixed and temporary hospitals in Fig. 1 is unclear, and the relief routes are not easy to distinguish between fixed and temporary hospitals.

Figure 1 was eliminated by decision of the work team.

Figure 1 was eliminated

Whether the objective function (formula (1)) is written in a standard way?

The objective function you mentioned was designed to reduce the suffering of people, so it makes sense to reduce time. The restrictions, on the other hand, were established to reduce the deprivation cost with a more efficient distribution of resources. By balancing these two objectives, a more effective and efficient disaster management plan can be achieved.

Lines 264 - 271

 The positions of Table 1 should be reformatted and should not be inserted between constraints.

The position of Table 1 have a new location.

The position of Table 1 have a new location.

Formula (7) mentions that the shelter will only be opened when the number of victims being cared for exceeds 10% of the shelter's capacity. What is the reason for selecting the 10% value?

The reason for selecting the value of 10% is that this value has been given by the relief agencies in the study area and has been validated by them. In addition, it considers the number of inhabitants and the response capacity for this type of disaster. This means that 10% is a value that is considered adequate for the area in question, considering the specific characteristics of the area.

Lines 298 - 299

Round 2

Reviewer 1 Report

In this new version, the authors addressed all my concerns in the previous version. However, the research has several pre-assumptions and limitations. I request the authors add a paragraph that discusses these limitations at the end of the conclusion section.

Author Response

Reviewer: 1

Reviewer #1's Comment

Answer to Reviewer

Implemented changes

However, the research has several pre-assumptions and limitations. I request the authors add a paragraph that discusses these limitations at the end of the conclusion section.

We added a paragraph in the 6. Conclusion and also a sentence in 3.1 Problem Definition

Lines 533 - 547

Lines 235 - 236

Reviewer 2 Report

1.      Please show more detail about how to use FlexSim software to solve the model. How to ensure the optimal solutions of the problem?

2.      Figure 3 should be improved, it is not clear to see as present.

Moderate editing of English language required

Author Response

Reviewer: 2

Reviewer #2's Comments

Answer to Reviewer

Implemented changes

1. Please show more detail about how to use FlexSim software to solve the model. How to ensure the optimal solutions of the problem?

We added a paragraph in the 3.3 Simulation Model Section

Lines 339-353

2. Figure 3 should be improved. It is not clear to see as present

The Figure has been redone. We also reviewed Figure 1.

Figure 3: Lines 366 – 368

Figure 1: Lines 228 – 230

Round 3

Reviewer 2 Report

The paper looks good now.

ok